

# Genome-wide identification and expression analysis of growth-regulating factors in *Dendrobium officinale* and *Dendrobium chrysotoxum*

Shuying Zhu[1,2,*], Hongman Wang[3,*], Qiqian Xue[3], Huasong Zou[1], Wei Liu[3], Qingyun Xue[3] and Xiao-Yu Ding[2,3]

[1] Huzhou College, School of Life and Health Sciences, Huzhou, Zhejiang, China
[2] Jiangsu Provincial Engineering Research Center for Technical Industrialization for Dendrobiums, Nanjing, Jiangsu, China
[3] Nanjing Normal University, College of Life Sciences, Nanjing, Jiangsu, China
[*] These authors contributed equally to this work.

Corresponding authors
Qingyun Xue, Qyxue1981@126.com
Xiao-Yu Ding, dingxynj@263.net

## ABSTRACT

**Background**. *Dendrobium*, one of the largest genera in Orchidaceae, is popular not only for its aesthetic appeal but for its significant medicinal value. Growth-regulating factors (GRFs) play an essential role in plant growth and development. However, there is still a lack of information about the evolution and biological function analysis of the *GRF* gene family among *Dendrobium* species.

**Methods**. Growth-regulating factors from *Dendrobium officinale* Kimura et Migo and *Dendrobium chrysotoxum* Lindl. were identified by HMMER and BLAST. Detailed bioinformatics analysis was conducted to explore the evolution and function of *GRF* gene family in *D. officinale* and *D. chrysotoxum* using genomic data, transcriptome data and qRT-PCR technology.

**Results**. Here, we evaluated the evolution of the *GRF* gene family based on the genome sequences of *D. officinale* and *D. chrysotoxum*. Inferred from phylogenetic trees, the *GRF* genes were classified into two clades, and each clade contains three subclades. Sequence comparison analysis revealed relatively conserved gene structures and motifs among members of the same subfamily, indicating a conserved evolution of *GRF* genes within *Dendrobium* species. However, considering the distribution of orthologous *DoGRFs* and *DcGRFs*, and the differences in the number of *GRFs* among species, we suggest that the *GRF* gene family has undergone different evolutionary processes. A total of 361 *cis*-elements were detected, with 33, 141, and 187 related to plant growth and development, stress, and hormones, respectively. The tissue-specific expression of *GRFs* showed that *DoGRF8* may have a significant function in the stem elongation of *D. officinale*. Moreover, four genes were up-regulated under Methyl-jasmonic acid/methyl jasmonate (MeJA) treatment, showing that *DoGRFs* and *DcGRFs* play a crucial role in stress response. These findings provide valuable information for further investigations into the evolution and function of *GRF* genes in *D. officinale* and *D. chrysotoxum*.

## INTRODUCTION

To adapt to changes in the growing environment, almost all plants have developed a variety of mechanisms and complex signal networks to ensure their growth and development during long-term evolution, and transcriptional regulation of gene expression is an important component. Transcription factors (TFs), which act as master regulators of gene expression, have an impact on the development of land plants, including the establishment of metabolism, species differentiation, and plant reproduction (*Shi et al., 2019*). The majority of TFs in plants are related to gene families such as MYB, WRKY, and TCP. Among them, growth-regulating factors (GRFs) play an important role in plants. It has been proven to be involved in the growth and development of multiple plant organs, particularly in stems and leaves. Initially, studies on *GRF*s mainly focused on their function in the development of plant leaves and stems (*Van der Knaap, Kim & Kende, 2000*; *Kim, Choi & Kende, 2003*; *Horiguchi, Kim & Tsukaya, 2005*; *Kim & Lee, 2006*). However, recent research has discovered their involvement in other aspects of plant growth and development, including seed and root development (*Bao et al., 2014*; *Debernardi et al., 2014*), growth control under stress conditions (*Pajoro et al., 2014*; *Liu et al., 2014*), and regulation of plant longevity (*Liang et al., 2014*; *Kim et al., 2012*; *Hewezi et al., 2012*). Therefore, *GRF*s play a crucial role in the growth and development of plants.

Previous research has identified two conserved domains located in the N-terminal portion of *GRF* genes: QLQ and WRC (*Van der Knaap, Kim & Kende, 2000*; *Omidbakhshfard et al., 2015*). The WRC domain, unique to plants, is expected to be involved in DNA binding and TF targeting to the nucleus. It can bind with the *cis*-acting region to regulate gene expression (*Choi, Kim & Kende, 2004*; *Zhang et al., 2008*). On the other hand, the QLQ domain serves as a protein-protein interaction domain and can interact with the *GRF*-interacting factor (GIF) family to form the *GRF-GIF* complex. This complex activates transcription and regulates plant growth and development. For instance, *AtGRF5* and *AtGIF1* cooperate to promote the development of leaf primordia (*Horiguchi, Kim & Tsukaya, 2005*).

With an increasing number of high-quality genome sequences of plant species being published, the *GRF* gene family has become popular in molecular evolution analyses. The *GRF* family has been identified in various species, including *Arabidopsis thaliana* (L.) Heynh. (*Kim, Choi & Kende, 2003*), *Brassica rapa* var. glabra Regel (*Wang et al., 2014*), *Zea mays* L. (*Zhang et al., 2008*), and *Oryza sativa* L. (*Choi, Kim & Kende, 2004*). *Dendrobium*s, as an endangered orchid, grows in adverse conditions, *e.g.*, epiphytic on cliffs or tree trunks, and distributed at high altitudes. Most of them have significant horticultural and medicinal values, such as *Dendrobium officinale* Kimura et Migo and *Dendrobium chrysotoxum* Lindl. (*Zhu et al., 2018*; *Li et al., 2020*; *Niu et al., 2018*). The stem of *D. officinale*, in particular, is a rare Chinese medicinal material with high market demand. *OsGRF1*, the first reported member of the *GRF* family, has been shown to regulate gibberellic acid-induced stem elongation and transcriptional activity (*Van der Knaap, Kim & Kende, 2000*). Therefore, it is crucial to understand the functions of *GRF*s in flowering, stem and leaf growth, seed formation, and root development in *Dendrobium*species. However, the evolution of the

*GRF* family among *Dendrobium* species is still unknown. With the recent availability of chromosome-level genome sequences for *D. officinale* and *D. chrysotoxum* (*Niu et al., 2021*; *Zhang et al., 2021*), it is now possible to conduct a comprehensive study of the *GRF* gene family in these species.

Therefore, in this study, we employed bioinformatics techniques to search for *GRF* genes using the genome sequences of *D. officinale* and *D. chrysotoxum* as references. We characterized their sequence attributes, chromosomal locations, evolutionary relationships, and conducted syntenic and gene duplication analyses. Additionally, we predicted *cis*-elements, expression patterns, 3D protein structures, and protein-protein interaction networks of the *GRF* genes to uncover their potential biological functions. These findings will provide valuable insights into the *GRF* gene family in both *Dendrobium* species and may pave the way for future research in this field.

# MATERIALS & METHODS

## Plant materials

The *D. officinale* and *D. chrysotoxum* used in this study were all from the well-growing rooting stage tissue culture seedlings in the *Dendrobiums* tissue culture Room, Institute of Plant and Environmental Resources, College of Life Sciences, Nanjing Normal University. *D. officinale* and *D. chrysotoxum* seedlings treated with 100 μM MeJA were used as the treatment group, and the seedlings with normal growth were used as the control group. After the treatment, the *D. officinale* and *D. chrysotoxum* seedlings were removed from the culture bottle, washed with water 2-3 times, and then absorbed with absorbent paper, and frozen in liquid nitrogen, and stored in an ultra-low temperature refrigerator at −80 °C for use.

## Identification of *GRF*s in *D. officinale* and *D. chrysotoxum* genome

First, we downloaded the HMM profiles of the *GRF* gene family (PF00244) from the Pfam protein family database (http://pfam-legacy.xfam.org/). Using these profiles, we conducted a search for candidate GRF proteins in the two *Dendrobium* species, with a parameter setting of *E*-value = 1e−5. Additionally, we obtained the *GRF* sequences of *A. thaliana* from the NCBI (https://www.ncbi.nlm.nih.gov/). These sequences were used in a BLASTP search to identify proteins in the two *Dendrobium* species. The protein sequences obtained from both methods were integrated to obtain putative *DoGRF*s and *DcGRF*s. To ensure the presence of conserved domains, these sequences were submitted to the SMART (http://smart.embl-heidelberg.de/), NCBI-CDD (https://www.ncbi.nlm.nih.gov/Structure/cdd/wrpsb.cgi), and Pfam websites. Finally, we utilized the ExPASy software online (https://web.expasy.org/protparam/, *Wilkins et al., 1999*) to analyze the features of *DoGRF*s and *DcGRF*s, including molecular weight, gene distribution, theoretical isoelectric point, and length. The subcellular localization was predicted using Cell-PLoc v2.0 software online (http://www.csbio.sjtu.edu.cn/bioinf/Cell-PLoc-2/).

## Phylogenetic trees, gene motifs and structures

First, the *GRF* amino acid sequences of *D. officinale*, *D. chrysotoxum*, *Phalaenopsis equestris* (Schauer) Rchb. (*Cai et al., 2015*), *Cymbidium ensifolium* (L.) Sw. (*Ai et*

*al., 2021*), and *A. thaliana* from NCBI (https://www.ncbi.nlm.nih.gov/) and NGDC (https://ngdc.cncb.ac.cn/) were aligned using MEGA7 (*Kumar, Stecher & Tamura, 2016*). The phylogenetic trees were constructed using the Neighbor-Joining method, with a bootstrap value of 1000, using the MEGA7 software. Next, we identified conserved motifs using the online MEME website (https://meme-suite.org/meme/, *Bailey & Elkan, 1994*), with a motif number of 10 and other parameters set to default. Additionally, we used the GSDS software online (http://gsds.gao-lab.org/index.php, *Hu et al., 2015*) to visualize the exon-intron structures of each sequence.

## Evolution analysis of gene duplications and collinearity within *Dendrobiums*

To start, we aligned the *DoGRF*s and *DcGRF*s using BLASTN with a parameter setting of *E*-value threshold = 1e−20 against the genome sequence of the two *Dendrobium*species. Next, based on the BLASTN results, we identified gene duplication events using MCScanX. The duplication events of *DoGRF*s and *DcGRF*s were visualized using the TBtools v1.6 software (*Wang et al., 2012*; *Chen et al., 2020*). Additionally, we determined the syntenic blocks between the two analyzed *Dendrobium*species and other plants using the MCScanX software, with the parameter of cscore ≥ 0.7.

## The calculation analysis of Ka and Ks

We used the software KaKs_Calculator v2.0 (*Wang et al., 2010*) to calculate the synonymous (*Ks*) value and non-synonymous (*Ka*) value. Additionally, we estimated the comparative ratio of *Ka* and *Ks*.

## Promoter analysis

The upstream 1,500 bp genomic DNA sequences of *GRF* genes were extracted as putative promoters. These promoters were then submitted to the PlantCare database (https://bioinformatics.psb.ugent.be/webtools/plantcare/html/, *Lescot et al., 2002*) for searching and analyzing the putative *cis*-elements. The total *cis*-elements were visualized using the TBtools software.

## Expression profiles

To investigate the different expression patterns of the *DoGRF*s, we conducted a search in the online database of NCBI SRA for RNA-sequence data from root, stem, leaf, and flower. The login IDs for the expression data are SRR2014476, SRR2014396, SRR2014325, SRR2014297, SRR2014230, SRR2014227, SRR1917043, SRR1917042, SRR1917041, and SRR1917040 (*Chen et al., 2017*). Firstly, the download RNA-sequence data were converted to fastq format *via* fastq-dump of SRA toolkit.3.0.0. Then the clean reads were aligned and mapped to the *D. officinale* genome by Hisat2 v2.2.1. The sam data was converted to bam by SAMtools v1.14. The FPKM value of *DoGRF*s were calculated by StringTie v2.2.0 (*Pertea et al., 2015*) to estimate the transcript abundances. To visualize the expression patterns, we constructed a heat map using the heatmap package in RStudio v1.4.1717 software (*RStudio Team, 2021*).

## Quantitative real-time PCR analysis of *DoGRFs* and *DcGRFs*

The extracted materials of RNA were reverse-transcribed by PrimeScript 1-strand cDNA synthesis kit (TaKaRa). Each reaction had a total volume of 20 μL, including SYBR Green I fluorescent dye 10 μL, primer (10 μM) 0.4 μL, cDNA 2 μL and ddH$_2$O 7.2 μL. The reaction conditions were predenaturation at 95 °C for 30s, 40 cycles (95 °C 5 s, 60 °C 30s), and dissolution curve (95 °C 15 s, 60 °C 60 s, 95 °C 15 s) (Tables S2–S4). We designed the primers using SnapGene v6.0 software (http://www.snapgene.com), and calculated the expression data using the method inferred from *Livak & Schmittgen (2002)*.

## The prediction of 3D protein structure and interaction network analysis

We predicted the 3D structures of GRF proteins from *D. officinale* and *D. chrysotoxum* using the online software SWISS-MODEL (https://swissmodel.expasy.org/, *Waterhouse et al., 2018*). First, we aligned the GRF protein sequences using the STRING v11.0 database online (https://cn.string-db.org/cgi/input?sessionId=bMUfhtTbeC2f{&}input_page_show_search=on) to predict their relationships, and the regulatory networks were visualized using the Gephi v0.9.6 software (*Von Mering et al., 2003*).

# RESULTS

## Identification and distribution of *GRFs* in *D. chrysotoxum* and *D. officinale*

A total of 37 *GRF* genes were identified from the genomes of *D. officinale* and *D. chrysotoxum*, with 19 and 18 *GRFs* identified using the methods of HMMER and BLASTP, respectively. There were differences in the characteristics of *GRF* genes between *D. officinale* and *D. chrysotoxum*. For example, the *DoGRF* proteins had a higher number of variable amino acids (ranging from 106 in *DoGRF16* to 392 in *DoGRF6*) compared to *DcGRF* proteins (ranging from 86 in *DcGRF8* to 321 in *DcGRF3*). The molecular weight of *DoGRF* proteins ranged from 11.7 kDa (*DoGRF16*) to 42.9 kDa (*DoGRF6*), which was higher than that of *DcGRF* proteins (ranging from 9.8 kDa in *DcGRF8* to 37.1 kDa in *DcGRF6*). Additionally, the isoelectric point of *DoGRF* proteins (ranging from 4.19 in *DoGRF16* to 10.07 in *DoGRF12*) was higher than that of *DcGRF* proteins (ranging from 4.02 in *DcGRF13* to 9.28 in *DcGRF10*).

The *DoGRFs* and *DcGRFs* were distributed on seven and eight chromosomes, respectively, among the 19 assembled chromosomes of *D. officinale* and *D. chrysotoxum*. As shown in Tables 1–2 below, most *DoGRFs* and *DcGRFs* were evenly distributed among the chromosomes mentioned. Notably, Chromosome 10 (Chr10) exhibited the highest number of *DcGRF* genes (Table 2). In addition, almost all the *GRF* genes from *D. officinale* and *D. chrysotoxum* were predicted to be distributed in nucleus and cytoplasm, which were probably the main working region for *GRF* genes.

## Phylogenetic analysis of *DoGRFs* and *DcGRFs*

The phylogenetic relationships are crucial for understanding the possible evolution of *DoGRFs* and *DcGRFs*. Using the Neighbor-Joining method, a phylogenetic tree was

**Table 1  The characteristics of *GRF* members identified in *Dendrobium officinale*.**

| No. | Gene name | Gene ID | Chr | Genomic location | Protein | Molecular weight (kDa) | Theoretical pI | Subcellular location |
|-----|-----------|---------|-----|------------------|---------|------------------------|----------------|----------------------|
| 1 | *DoGRF1* | *Dof000773* | 1 | 25271270-25289574 | 246 | 27.782 | 4.81 | N. |
| 2 | *DoGRF2* | *Dof001775* | 1 | 87182419-87197524 | 275 | 31.231 | 4.59 | N. |
| 3 | *DoGRF3* | *Dof007872* | 5 | 2696374-2767924 | 258 | 28.952 | 4.43 | N. |
| 4 | *DoGRF4* | *Dof007881* | 5 | 2922050-2950591 | 251 | 28.255 | 4.73 | N. |
| 5 | *DoGRF5* | *Dof011242* | 7 | 3937557-3941920 | 262 | 29.622 | 4.6 | N. |
| 6 | *DoGRF6* | *Dof011366* | 7 | 8699540-8742219 | 392 | 42.960 | 4.59 | C. |
| 7 | *DoGRF7* | *Dof011962* | 7 | 63779975-63798763 | 258 | 29.101 | 4.53 | N. |
| 8 | *DoGRF8* | *Dof014759* | 10 | 7700573-7705243 | 258 | 29.057 | 4.48 | N. |
| 9 | *DoGRF9* | *Dof014810* | 10 | 9466121-9476099 | 290 | 32.711 | 4.46 | N. |
| 10 | *DoGRF10* | *Dof016970* | 12 | 19189885-19205441 | 355 | 38.987 | 9.96 | C. N. |
| 11 | *DoGRF11* | *Dof016971* | 12 | 19206077-19241077 | 372 | 40.313 | 8.57 | C. |
| 12 | *DoGRF12* | *Dof016973* | 12 | 19402650-19434919 | 301 | 33.371 | 10.07 | C. N. |
| 13 | *DoGRF13* | *Dof021876* | 16 | 1828252-1846553 | 258 | 28.950 | 4.55 | N. |
| 14 | *DoGRF14* | *Dof021877* | 16 | 1848488-1849460 | 243 | 27.259 | 6.92 | C. N. |
| 15 | *DoGRF15* | *Dof022251* | 16 | 11590340-11593140 | 256 | 28.766 | 4.48 | N. |
| 16 | *DoGRF16* | *Dof023056* | 17 | 7309376-7318548 | 106 | 11.719 | 4.19 | N. |
| 17 | *DoGRF17* | *Dof023057* | 17 | 7318691-7321768 | 117 | 13.039 | 4.3 | N. |
| 18 | *DoGRF18* | *Dof023549* | 17 | 34990936-35016843 | 254 | 28.434 | 6.26 | C. N. |
| 19 | *DoGRF19* | *Dof026766* | UN | 158494-162784 | 257 | 29.258 | 5.1 | N. |

**Notes.**
*N*, nucleus; *C*, cytoplasm.

constructed with a total of 81 *GRF* genes from five species: *A. thaliana* (13), *D. officinale* (17), *D. chrysotoxum* (16), *Cymbidium ensifolium* (L.) Sw. (12), and *Phalaenopsis equestris* (Schauer) Rchb. (23) using MEGA software. The results showed that the 81 *GRF*s were divided into two major clades, designated as clade I and clade II (Fig. 1). Clade I was further subdivided into three subclades, labeled as A, B, and C, containing 8, 6, and 8 *GRF* genes, respectively. Clade II was also divided into three subclades, labeled as D, E, and F, containing 25, 18, and 16 *GRF* genes, respectively. Within different subclades, most of the *GRF* genes from the two *Dendrobium*species clustered together. Notably, we identified 8 pairs of orthologous genes with a close relationship (Bootstrap value >90), such as *DoGRF14* and *DcGRF12*. This finding suggests a close relationship between the *GRF*s of *Dendrobium*species. Furthermore, subfamilies A and B did not contain any *AtGRF* s or *CeGRF* s, whereas each of the other subfamilies included *GRF* genes from all five species. The number of *GRF* genes in different branches of closely related species was relatively consistent. Clade I included seven *GRF* genes from *D. officinale* and five *GRF* genes from *D. chrysotoxum*, respectively. Clade II included 10 *GRF* genes from *D. officinale* and 11 *GRF* genes from *D. chrysotoxum*, respectively. Considering the distribution of orthologous genes between *D. officinale* and *D. chrysotoxum* and the differences in the number of *GRF* genes among *Dendrobium*species, we suggest that while the *GRF* gene family has undergone

**Table 2** The characteristics of *GRF* members identified in *Dendrobium chrysotoxum*.

| No. | Gene name | Gene ID | Chr | Genomic location | Protein | Molecular weight (kDa) | Theoretical pI | Subcellular location |
|---|---|---|---|---|---|---|---|---|
| 1 | *DcGRF1* | KAH0449449 | 18 | 27696253-27735081 | 271 | 30.820 | 4.72 | N. |
| 2 | *DcGRF2* | KAH0449492 | 18 | 90325059-90331686 | 260 | 29.437 | 4.46 | N. |
| 3 | *DcGRF3* | KAH0453121 | 16 | 4317217-4350471 | 321 | 36.097 | 5.49 | N. |
| 4 | *DcGRF4* | KAH0453427 | 16 | 4024734-4056002 | 257 | 28.954 | 4.43 | N. |
| 5 | *DcGRF5* | KAH0453534 | 16 | 13763326-13768257 | 263 | 29.980 | 4.9 | N. |
| 6 | *DcGRF6* | KAH0455432 | 14 | 36418564-36435531 | 320 | 37.198 | 8.05 | C. N. |
| 7 | *DcGRF7* | KAH0455781 | 14 | 5996726-6009621 | 258 | 29.122 | 4.24 | N. |
| 8 | *DcGRF8* | KAH0458321 | 12 | 1245507-1245858 | 86 | 9.887 | 4.41 | C. M. N. |
| 9 | *DcGRF9* | KAH0458964 | 11 | 18249753-18250544 | 263 | 28.647 | 8.81 | C. N. |
| 10 | *DcGRF10* | KAH0459081 | 11 | 18342320-18343057 | 245 | 27.582 | 9.28 | C. N. |
| 11 | *DcGRF11* | KAH0459817 | 10 | 14861543-14863463 | 255 | 28.766 | 4.48 | N. |
| 12 | *DcGRF12* | KAH0459925 | 10 | 2214094-2217553 | 264 | 29.288 | 4.9 | C. N. |
| 13 | *DcGRF13* | KAH0460355 | 10 | 2202923-2212464 | 142 | 16.021 | 4.02 | N. |
| 14 | *DcGRF14* | KAH0460481 | 10 | 2187485-2192245 | 257 | 29.302 | 5.32 | N. |
| 15 | *DcGRF15* | KAH0464062 | 7 | 48667464-48672171 | 257 | 29.057 | 4.48 | N. |
| 16 | *DcGRF16* | KAH0464557 | 7 | 46549349-46565636 | 265 | 30.168 | 4.74 | N. |
| 17 | *DcGRF17* | KAH0468224 | 4 | 70550448-70567786 | 257 | 29.118 | 4.53 | N. |
| 18 | *DcGRF18* | KAH0468498 | 4 | 4229841-4241919 | 261 | 29.606 | 4.6 | N. |

**Notes.**

*M*, microbody; *N*, nucleus; *C*, cytoplasm.

different evolutionary processes (gene loss or gain), the evolution of *GRF* genes remains conservative in closely related species.

## Gene motifs and structures of *DoGRF*s and *DcGRF*s

The amino acid sequences of 17 *DoGRF*s and 16 *DcGRF*s were used to construct phylogenetic trees. To further analyze their motif compositions, these sequences were submitted to the MEME website. The results revealed that both *D. officinale* and *D. chrysotoxum* exhibited 10 motifs within a length range of 14aa-50aa. However, upon examining the detailed sequence information, differences in motifs between the two species were observed. Figure 2 shows that motifs 1-7 were widely distributed in the majority of *DoGRF*s, while motifs 8-10 were found in only three genes. Similar distribution patterns were observed in *D. chrysotoxum*, with motifs 1-7 being relatively conserved and widespread among most *DcGRF*s, except for *DcGRF6*, which exhibited a unique distribution pattern with only 1 motif (Fig. 2 and Fig. S1).

By referring to published genomic information, the structure of *GRF*s was further elucidated through exon-intron structure analysis. The results demonstrated that *GRF*s within the same species shared a highly similar structure. The lengths and numbers of exons clustered together in the phylogenetic tree were nearly identical, and the lengths of introns were also highly similar. This indicates that *GRF*s in both species have been evolutionarily

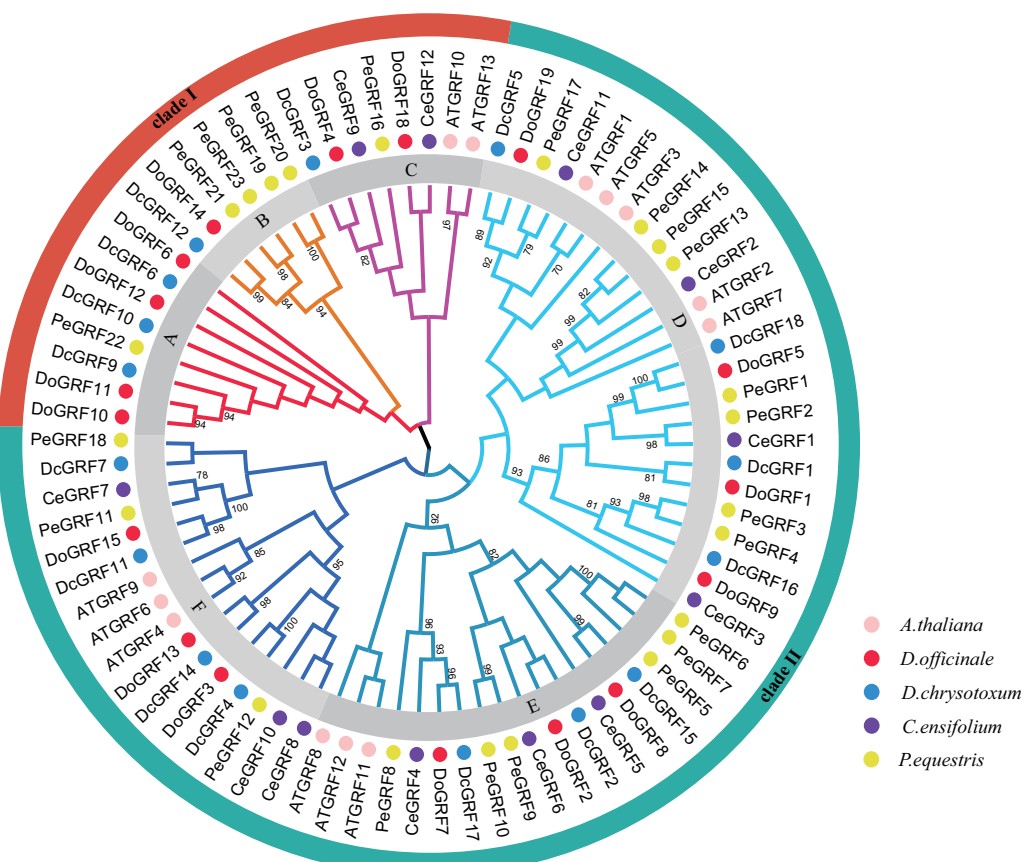

**Figure 1** **Phylogenetic relationships of *GRF* genes in *D. officinale*, *D. chrysotoxum*, *A. thaliana*, *C. ensifolium* and *P. equestris*.** Neighbor-Joining phylogenetic tree was constructed by MEGA7 with 1000 bootstraps. Pink, red, blue, purple and yellow colors represent GRF protein sequences from *A. thaliana* (AT), *D. officinale* (Do), *D. chrysotoxum* (Dc), *C. ensifolium* (Ce) and *P. equestris* (Pe), respectively. Different subfamilies are shaded with different colors.

conserved. However, compared to *D. officinale*, *D. chrysotoxum* had slightly fewer introns in its *GRF*s.

## Gene duplication of *DoGRF*s and *DcGRF*s

To investigate *GRF* gene duplication events and uncover potential evolutionary histories in *D. officinale* and *D. chrysotoxum*, BLASTN and MCScanX were employed for synteny analysis of *GRF* genes between the two species. The results revealed the presence of similar homologous gene pairs (eight pairs in *D. officinale* and nine pairs in *D. chrysotoxum*, as shown in Fig. 3) and approximate replication patterns. Detailed examination showed that segmental duplications were widely distributed in both species, with a clear predominance, while tandem duplications were also observed in one gene pair in both *D. officinale* and *D.*

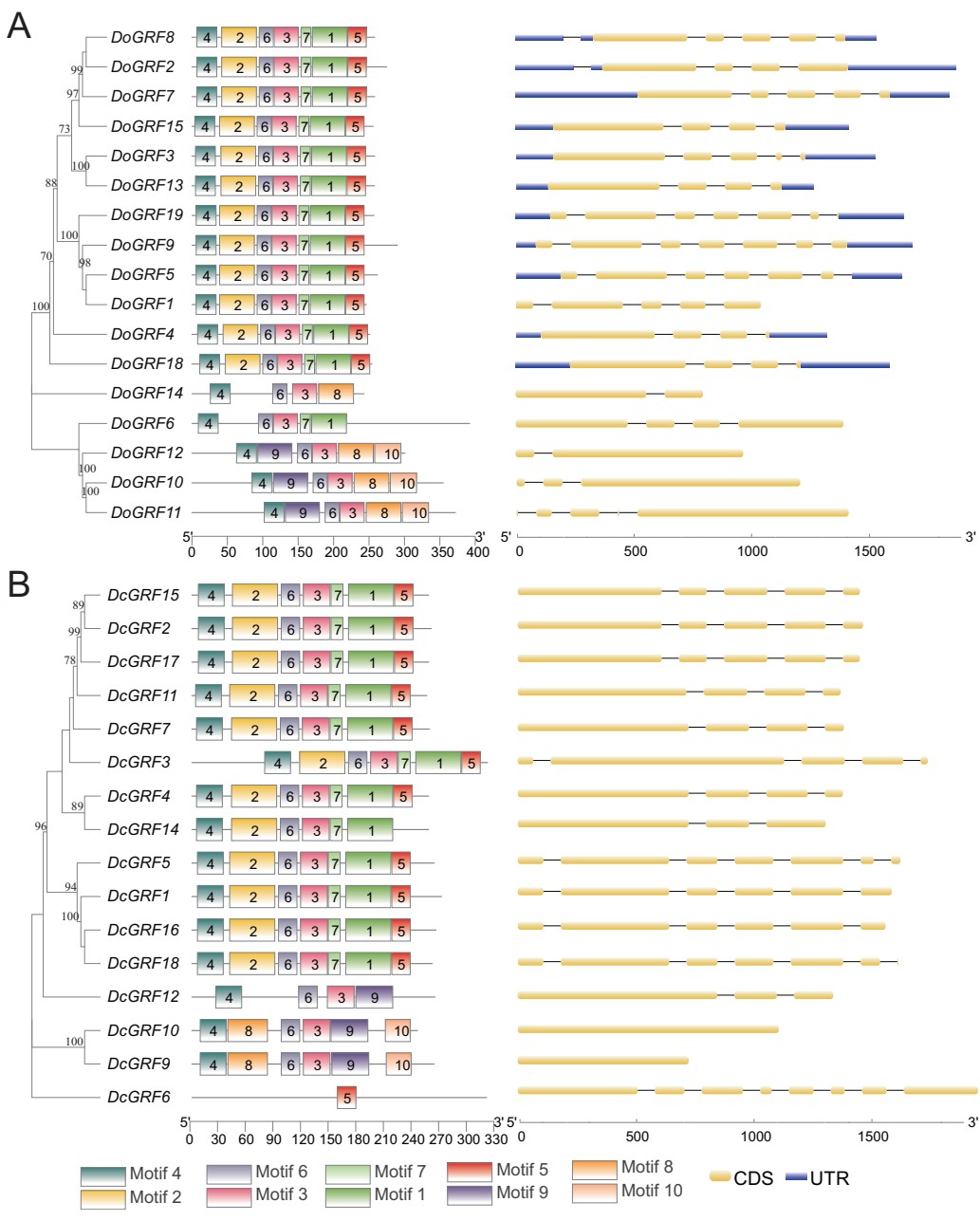

**Figure 2** Phylogenetic relationships, conserved motifs and exon-intron structures of *GRF* genes in *D. officinale* (A) and *D. chrysotoxum* (B). The conserved motifs were identified using MEME and visualized by TBtools. Different colors represent 10 different motifs. Yellow and blue boxes are respectively indicating CDS and UTR.

*chrysotoxum*. Therefore, segmental duplication was the main mechanism contributing to the expansion of *DoGRF*s and *DcGRF*s.

The *Ka/Ks* ratio, which measures the frequency of non-synonymous (*Ka*) and synonymous (*Ks*) substitutions in homologous pairs of *DoGRF*s and *DcGRF*s, was used to

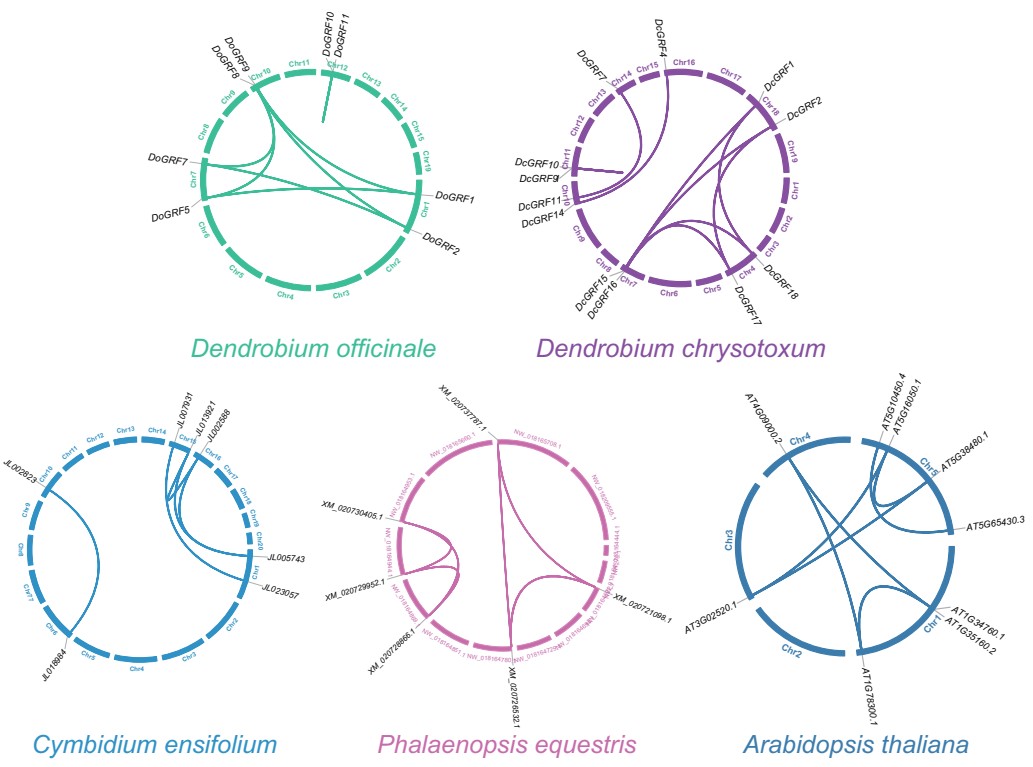

**Figure 3** Schematic representations of the gene duplications of *GRF* genes from five different plants.

assess the presence of selection pressure. Among the eight pairs of *D. officinale GRF* genes, seven pairs exhibited purifying selection effects, while one pair had a *Ka/Ks* ratio greater than one, indicating positive selection effects (Table 3).

In addition, we compared the replication events between two *Dendrobium*s and other three species (*A. thaliana*, *C. ensifolium*, and *P. equestris*) to further understand the replication event of *GRF*s. A total of five, six and seven paralogous genes were detected among *C. ensifolium*, *P. equestris* and *A. thaliana*, respectively. Among these paralogs, all the gene pairs experienced a negative selection (Ka/Ks <1), which were conserved (Fig. 3 and Table S5).

## Syntenic analysis of *DoGRF*s and *DcGRF*s

Interspecific collinearity analysis provides valuable insights into the evolution of gene families. We conducted collinearity analysis between *DoGRF*s and *DcGRF*s, and further examined their collinear relationships with *A. thaliana*, *O. sativa*, and *Vanilla planifolia* Andrews, as depicted in Fig. 4. (i) Collinear analysis revealed that *D. officinale* and *D. chrysotoxum* exhibited the highest number of homologous genes, with a total of 12 gene pairs. Specifically, there were five pairs of homology genes between *D. officinale* and *O. sativa*, nine pairs of homology genes between *D. officinale* and *V. planifolia*, and only four pairs of homology genes between *D. officinale* and *A. thaliana*. Similarly, there were five pairs of homology genes between *D. chrysotoxum* and *O. sativa*, seven pairs of

**Table 3  Ka, Ks and Ka/Ks values for duplication gene pairs in *DoGRFs* and *DcGRFs*.**

| Seq_1 | Seq_2 | Ka | Ks | Ka/Ks | Duplication type |
|---|---|---|---|---|---|
| *DoGRF1* | *DoGRF5* | 0.04546 | 0.959153 | 0.047396 | Segmental duplication |
| *DoGRF1* | *DoGRF9* | 0.062838 | 2.52223 | 0.024914 | Segmental duplication |
| *DoGRF2* | *DoGRF7* | 0.987025 | 1.03655 | 0.952224 | Segmental duplication |
| *DoGRF2* | *DoGRF8* | 0.059869 | 0.928183 | 0.064502 | Segmental duplication |
| *DoGRF3* | *DoGRF13* | 0.061013 | 1.01233 | 0.06027 | Segmental duplication |
| *DoGRF5* | *DoGRF9* | 0.978744 | 1.07199 | 0.913013 | Segmental duplication |
| *DoGRF7* | *DoGRF8* | 0.052936 | 2.86335 | 0.018487 | Segmental duplication |
| *DoGRF10* | *DoGRF11* | 0.108897 | 0.102434 | 1.06309 | Tandem duplication |
| *DcGRF11* | *DcGRF7* | 0.0924322 | 1.43463 | 0.0644291 | Segmental duplication |
| *DcGRF18* | *DcGRF1* | 0.0781159 | 1.08391 | 0.0720686 | Segmental duplication |
| *DcGRF18* | *DcGRF16* | 0.0735203 | 3.65543 | 0.0201126 | Segmental duplication |
| *DcGRF9* | *DcGRF10* | 0.994268 | 1.0153 | 0.979283 | Tandem duplication |
| *DcGRF15* | *DcGRF2* | 0.0285821 | 0.935346 | 0.0305577 | Segmental duplication |
| *DcGRF16* | *DcGRF1* | 0.0743845 | 2.03182 | 0.0366098 | Segmental duplication |
| *DcGRF14* | *DcGRF4* | 0.217967 | 1.44347 | 0.151002 | Segmental duplication |
| *DcGRF17* | *DcGRF15* | 0.0560215 | 2.47396 | 0.0226445 | Segmental duplication |
| *DcGRF17* | *DcGRF2* | 0.0501976 | 1.67991 | 0.0298811 | Segmental duplication |

**Notes.**
Synonymous (Ks) and nonsynonymous (Ka) substitution rates of duplicate gene pairs (Ka/Ks ratios).

homology genes between *D. chrysotoxum* and *V. planifolia*, and five pairs of homology genes between *D. chrysotoxum* and *A. thaliana*. The results indicate that the *GRF* gene families of monocots and dicots, such as *O. sativa* and *A. thaliana*, show relatively fewer differences, while more collinear relationships are observed among orchids. (ii) *DoGRF13* in *D. officinale* and *DcGRF3* in *D. chrysotoxum* exhibited homologous genes with the other four plants, suggesting a common ancestor predating the divergence of monocots and dicots and indicating functional conservation and importance. Excluding the influence of the dicot *A. thaliana*, it was observed that *DoGRF7*, *DoGRF2*, and *DoGRF8* in *D. officinale*, as well as *DcGRF17*, *DcGRF2*, and *DcGRF15* in *D. chrysotoxum*, displayed homologous genes in the other three monocots, suggesting relative conservation in monocot evolution. Furthermore, these six genes corresponded to collinear results between *D. officinale* and *D. chrysotoxum* (*DcGRF17-DoGRF7*, *DcGRF2-DoGRF2*, *DcGRF15-DoGRF8*). (iii) Compared to *O. sativa* and *A. thaliana*, orchids, including *D. officinale* and *D. chrysotoxum*, exhibited a significant doubling in the number of *GRF* genes. For instance, the gene *LOC_Os08g33370* in *O. sativa* displayed collinearity with three genes (*DoGRF7*, *DoGRF2*, *DoGRF8*) in *D. officinale* and two genes (*DcGRF17*, *DcGRF2*) in *D. chrysotoxum*, and numerous similar cases were observed. Additionally, even within the orchid family, *D. officinale* and *D. chrysotoxum* exhibited a doubling compared to vanilla orchid. For example, the gene *Vpl04Ag09642* in *V. planifolia* displayed collinearity with two genes (*DoGRF8*, *DoGRF2*) in *D. officinale*.

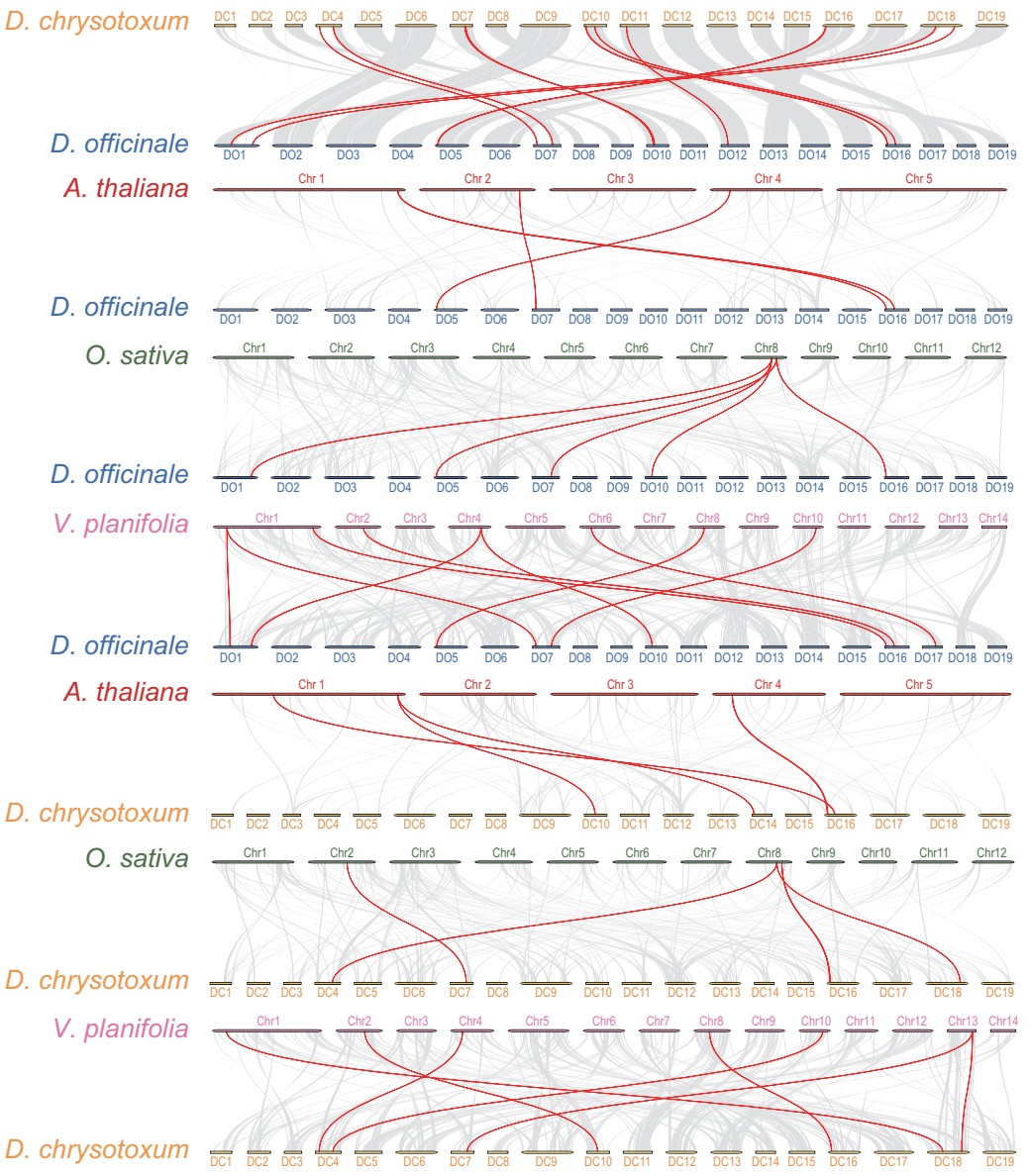

**Figure 4   Collinearity analysis of *GRF* genes in *D. officinale*, *D. chrysotoxum* and three other plants, including *A. thaliana*, *O. sativa* and *V. planifolia*.** Grey lines indicate the collinear blocks. Red lines indicate the collinear blocks of *GRF* genes.

## Analysis of *DoGRF*s and *DcGRF*s promoter

To gain a better understanding of the potential functions of *DoGRF*s and *DcGRF*s, we identified *cis*-elements within the 1,500 bp upstream regions of the initiation codon (ATG). After excluding non-functional terms, a total of 361 *cis*-elements in the promoter regions of *DoGRF*s and *DcGRF*s were categorized into three groups: plant development-related (9%), stress-responsive (39%), and hormone-related (52%).

Within the plant growth and development category (33/361), we identified five *cis*-elements involved in endosperm expression (GCN4-motif), cell cycle regulation (MSA-like), meristem expression (CAT-box), circadian control (circadian), and zein metabolism regulation (O2-site), with CAT-box accounting for the largest proportion.

In the stress responsiveness category (141/361), we identified *cis*-elements responsive to light (ACE, G-box, GT1-motif, and Sp1), low-temperature (LTR), defense and stress (TC-rich repeats), and anaerobic induction (ARE). Additionally, more than half of the *cis*-elements (187/361) were related to phytohormones, responding to various phytohormones such as ABA, auxin, GA, MeJA, and salicylic acid. Notably, MeJA-responsive and light-responsive *cis*-elements were the most abundant in both *D. officinale* and *D. chrysotoxum*.

These results suggest that MeJA-induced or suppressed *GRF* genes, along with those responding to various abiotic stresses, may play a role in photosynthesis (Fig. 5).

## Expression patterns of *GRF*s in different tissues and under MeJA treatments

To investigate the potential biological functions of *GRF*s in *D. officinale*, we analyzed the tissue-specific expression of *DoGRF*s using transcriptome data and created a heat map (Fig. 6A) based on FPKM values from roots, leaves, flowers, and stems at four different growth stages of *D. officinale*. The heat map revealed that more than half of the *DoGRF*s were expressed in stems, flowers, and leaves, but not in roots of *D. officinale*. Different expression patterns were observed in the four stages of stem development, with most genes showing the highest expression at 4 months. Previous studies have associated *GRF*s with stem elongation (*Van der Knaap, Kim & Kende, 2000*). Additionally, *cis*-acting element analysis showed that the CAT-box, related to stem and root meristem expression, accounted for the highest proportion of growth and development-related elements. Considering the presence of gibberellin-related elements and CAT-box, it can be speculated that *DoGRF8* may play a significant role in stem elongation in *D. officinale*.

Furthermore, *cis*-element analysis revealed a significant number of MeJA response elements within *DoGRF*s and *DcGRF*s. To explore the potential biological functions of *GRF*s under MeJA treatment, we selected 10 *DoGRF*s and 10 *DcGRF*s based on the expression results mentioned above and determined their expression levels using qRT-PCR (Fig. 6B). Among them, four genes were up-regulated, ten were down-regulated, and the remaining *GRF*s showed no significant changes in expression levels. These results suggest that MeJA treatment may affect the proper functioning of *GRF* genes in *Dendrobium*s.

## 3D structure prediction of GRF proteins of *D. officinale* and *D. chrysotoxum*

To explore the effect of protein structure on function, 19 *DoGRF*s and 18 *DcGRF*s were submitted to the SWISS-MODEL website for protein 3D structure prediction. Ultimately, 24 high-quality models with more than 30% consistency were generated (Table S1). The QMEAN DisCo Global value and GMQE value provided by the SWISS-MODEL website serve as quality evaluation standards. The QMEAN DisCo Global values of *DoGRF*s ranged from 0.76 to 0.87, and the GMQE values ranged from 0.72 to 0.88. The QMEAN DisCo Global values of *DcGRF*s ranged from 0.80 to 0.87, and the GMQE values ranged from

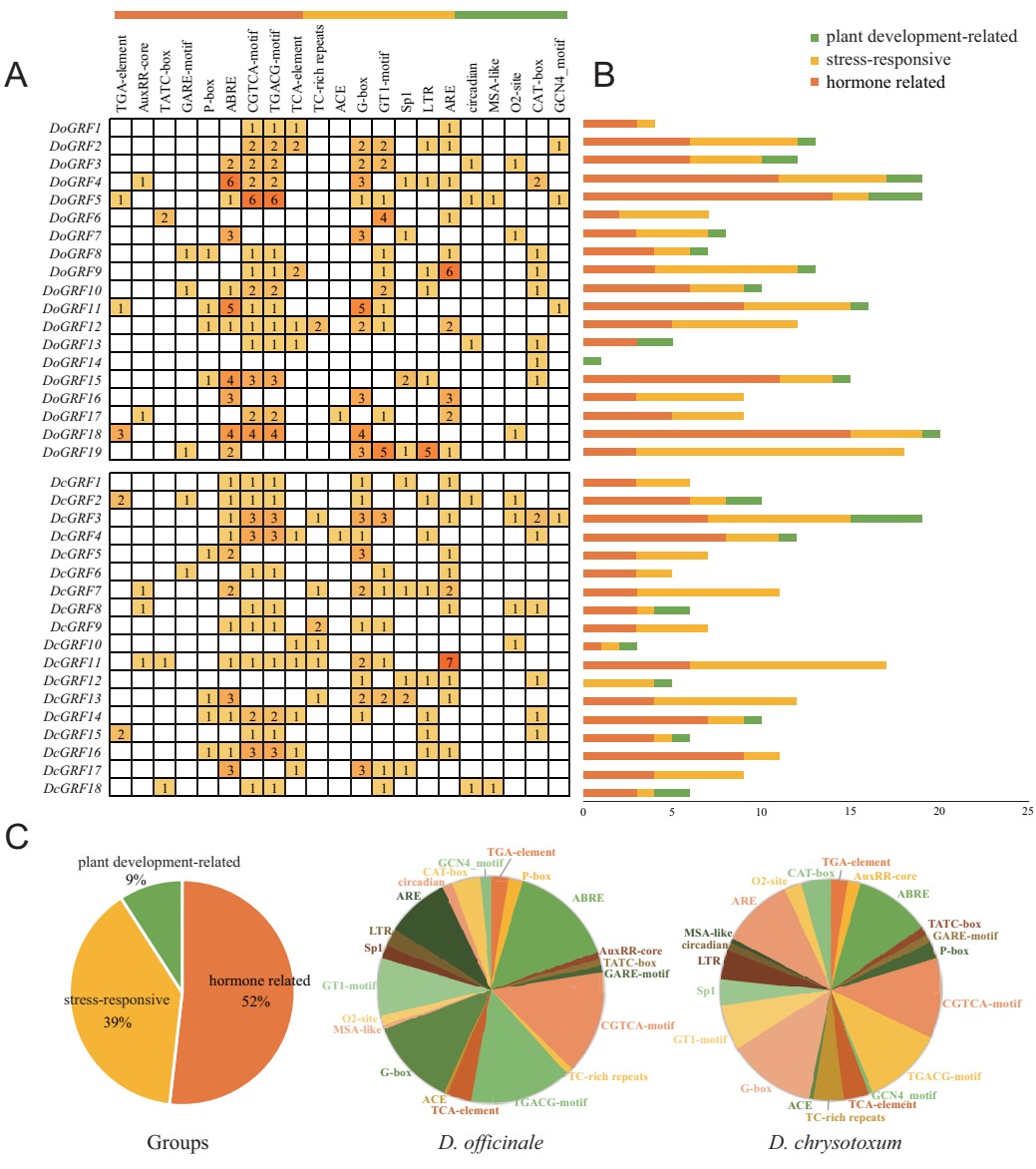

**Figure 5** **Information of *cis*-acting elements in *GRF* genes of *D. officinale* and *D. chrysotoxum*.** (A) The gradient orange colors and numbers in the grid indicate the number of different *cis*-elements. (B) The different colors histogram indicates the number of *cis*-elements in each category. (C) The ratio of different *cis*-acting elements in *D. officinale* and *D. chrysotoxum* is shown as pie charts.

0.61 to 0.88. Overall, the models exhibited good quality. Detailed data can be found in the attached table.

All 24 constructed models were Hom-Dimer Oligo-State, indicating a relatively conserved function. In both *D. officinale* and *D. chrysotoxum*, the *GRF* gene family exhibited two different protein structures due to variations in rotation angles. Similar protein structures are likely to have similar functions, while different protein structures may contribute to the functional diversity of *GRFs* in *Dendrobium*s.
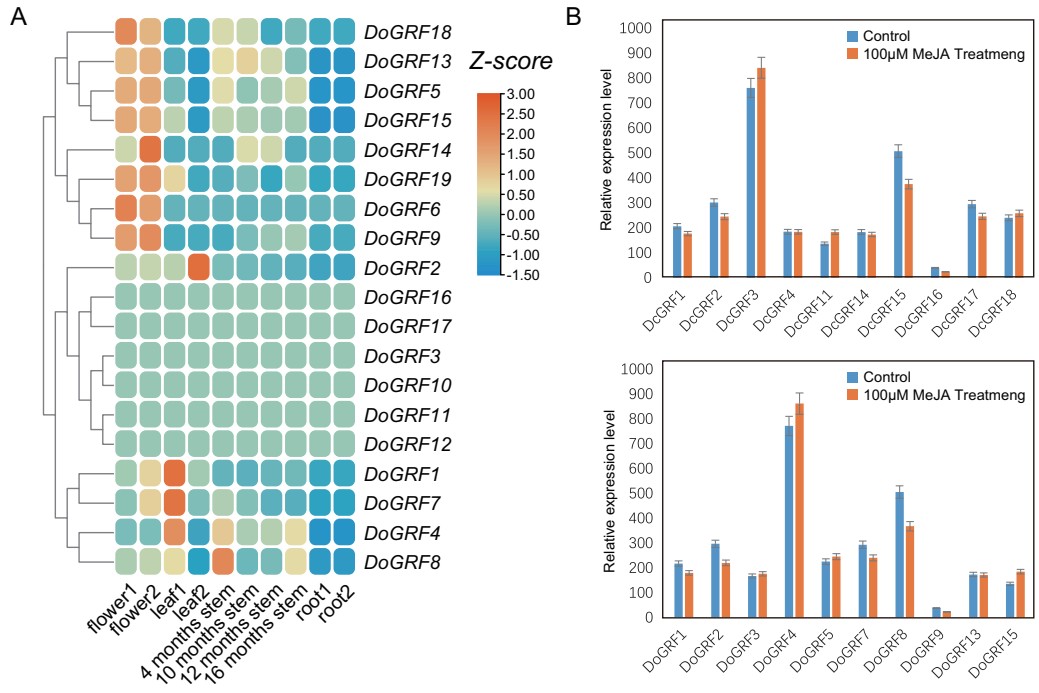

**Figure 6** **Expression analysis of *GRF* s in different tissues and MeJA treat.** (A) Expression profiles of *GRF* genes of *D. officinale* in different tissues including root, stem, leaf and flower. Z-score transformed FPKM values. (B) Relative expression levels of *DoGRF* s and *DcGRF* s under MeJA treatments.

## Protein-protein interaction networks of *DoGRF*s and *DcGRF*s

In order to gain a better understanding of the potential biological functions and regulatory networks of *GRF* genes, we predicted and constructed interaction networks between *GRF* proteins and related proteins in *D. officinale* and *D. chrysotoxum*, respectively. Our findings revealed complete consistency in the interactions between related proteins of both species, identifying a total of 65 related proteins and 233 connections. Among them, *DoGRF18* protein interacted with 43 proteins, while *DoGRF18* protein interacted with 38 proteins (including *GRF* proteins and related proteins), suggesting their involvement in multiple biological processes. On the other hand, five *GRF* proteins did not show any connections to related proteins. Additionally, based on homology and co-expression analysis, *DoGRF12*, *DoGRF16*, *DoGRF17*, and PBSO (oxygen-evolving enhancer protein 1, chloroplastic) exhibited the closest interaction relationship, with CMDH (Malate dehydrogenase) also being among the related proteins. PBSO and CMDH are known to play essential roles in photosynthesis. Hence, our results indicate that plant growth and development, encompassing multiple processes, may represent the most significant function of *DoGRF*s and *DcGRF*s (refer to Fig. 7 for details).
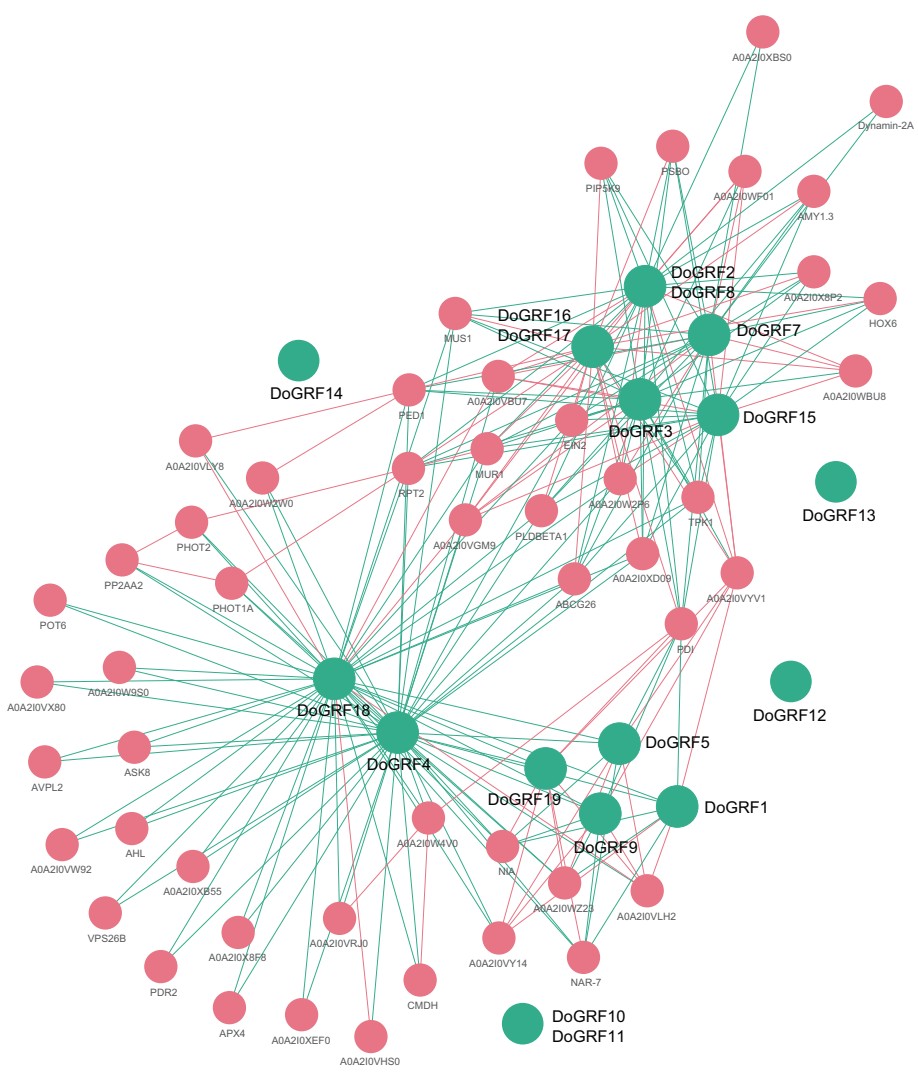

**Figure 7** **Protein-protein interaction (PPI) networks of GRF proteins in *D. officinale*.** Green and pink circles represent GRF and related proteins, respectively.

# DISCUSSION

## The evolution of *GRF*s is conserved within *Dendrobium* genus

The *GRF* family, a group of small transcription factors, plays crucial roles in various plant biological processes, including phytohormone responses, regulation of growth and development, and stress responses (*Vercruyssen et al., 2015*). For instance, *Hewezi et al. (2012)* focused on the study of *GRF*s in *A. thaliana* and found that highly expressed *AtGRF1* and *AtGRF3* in roots had a balanced expression that affected root growth. Gibberellin treatment, as a plant hormone, has been shown to increase the expression of several *GRF*s in rice and *B. rapa*. Additionally, *AtGRF7* mutants exhibit greater tolerance to drought and salinity stress compared to wild-type and *AtGRF7* overexpressor lines (*Liu et al., 1998*; *Kim et al., 2012*). While the genome-wide identification of *GRF*s has been reported in various

plant species, such as nine genes in *A. thaliana*, 13 genes in *O. sativa*, and 17 genes in *Z. mays* (*Kim, Choi & Kende, 2003*; *Choi, Kim & Kende, 2004*; *Zhang et al., 2008*), studies on the evolution and function of *GRF*s in *Dendrobium*species are still lacking despite the availability of high-quality *D. officinale* and *D. chrysotoxum* genome sequences.

The *GRF* family has been documented to undergo significant expansions/contractions among different plant lineages. For example, there are a total of nine and nine *GRF* genes in *Vitis vinifera* and *A. thaliana*, respectively (*Hu et al., 2023*; *Kim, Choi & Kende, 2003*), while 17 genes are found in *B. rapa* (*Wang et al., 2014*). On the contrary, *Z. mays* and *Gossypium raimondii* have 17 and 19 *GRF* genes, respectively (*Zhang et al., 2008*; *Cao et al., 2020*), whereas *Sorghum bicolor* has eight genes (*Shi et al., 2022*). Comparative analysis reveals significant expansion/contraction events among these species. In our study, we identified 19 and 18 *GRF*s in *D. officinale* and *D. chrysotoxum*, respectively. Although the gene numbers of *GRF*s vary between *Dendrobium* orchids and *A. thaliana*, *P. equestris*, and *C. ensifolium*, the evolution of the *GRF* gene family remains conserved within the genus of *Dendrobium*. For example, (i) the *GRF* genes among *Dendrobium* species have formed eight pairs of orthologous genes, which account for 43% of the total *GRF* genes. Importantly, we identified a pair of positively selected genes (*DoGRF10* and *DoGRF11*), suggesting that *DoGRF*s have undergone positive selection pressure. These findings directly demonstrate the conservation of *GRF* evolution among *Dendrobium*species. (ii) Collinearity analysis suggests that the *GRF* genes have experienced both expansion and contraction events in other plant lineages, but the most abundant homologous genes are found between *D. officinale* and *D. chrysotoxum*. (iii) A total of 17 gene duplications, with 8 and 9 repeats, were identified in *D. officinale* and *D. chrysotoxum*, respectively, indicating that gene duplication has been a driving force for *GRF* gene evolution, leading to a conserved gene family among *Dendrobium* orchids.

### The *GRF* gene family are important for plant development, stress response and hormone response among *Dendrobium* species

The *GRF* genes are members of an important plant-specific family that have been studied for their crucial role in central developmental processes in plants, including stem and leaf development, seed formation, flowering, and root development. For example, *AtGRF4* of *A. thaliana* has been reported to have various functions, such as cell proliferation in leaves, the shoot meristemless/*stm* mutant phenotype, and embryonic development of cotyledons (*Kim & Lee, 2006*; *Gonzalez, Beemster & Inzé, 2009*). *Pajoro et al. (2014)* revealed the role of *miR396a* in flower formation in *A. thaliana*, where it regulates *GRF* transcript levels and determines sepal-petal identity. Additionally, a regulatory network involving *miR396* and its targets, including *bHLH74* and *GRF*s, plays a central role in normal root growth and development (*Debernardi et al., 2012*; *Bao et al., 2014*). Recent research has highlighted the significant effects of *GRF* genes in photosynthesis, phytohormone signaling, and growth under adverse environmental conditions. For instance, (1) *AtGRF5* stimulates chloroplast division, leading to an increase in the number of chloroplasts per cell in *35S:GRF5* leaves and a consequent increase in chlorophyll levels, thereby maintaining a higher rate of photosynthesis (*Vercruyssen et al., 2015*). (2) *Van der Knaap, Kim & Kende (2000)* first

reported that the *GRF* member *OsGRF1* regulates GA3-induced stem elongation and transcriptional activity (*Kim & Kende, 2004*). (3) Further functional classification of the putative downstream targets of *AtGRF1* and *AtGRF3* has revealed that most of them are involved in defense responses and disease resistance processes (*Liu et al., 2014*).

Consistently, our results confirm that *GRF* genes have diverse biological functions related to plant development, stress response, and hormone signaling. For example, (i) *GRF*s play an important role in plant development. In our study, we detected 33 *cis*-elements involved in plant development, accounting for 9.14% of all predicted *cis*-elements. For instance, the expression of *DoGRF8* was closely related to stem development in *D. officinale*. Similar results were observed for *DoGRF1*, *DoGRF2*, *DoGRF7*, and *DoGRF14*, which were related to the development of flowers and leaves. (ii) As epiphytes growing at high altitudes above 800m, *Dendrobium*species have developed mechanisms to accumulate anti-stress substances, enhancing their ability to respond to harsh environments. In our study, we identified 141 *cis*-elements involved in stress response, accounting for 39.06% of all detected *cis*-elements. Moreover, based on our analysis of MeJA treatment, we found that *DoGRF4* and *DoGRF15*, which have been documented in *D. officinale*, were up-regulated, indicating their enhanced function in stress response in harsh habitats. (iii) We identified a total of 46 and 39 *cis*-elements involved in light responsiveness in *D. officinale* (23.12%) and *D. chrysotoxum* (24.07%), respectively, which may be related to the special photosynthetic pathway of *Dendrobium*s.

## The biological function of *GRF* gene family were closely related to the protein structure, gene evolution or duplication events and protein interaction

As reported by *Wang et al. (2022)*, different gene families exhibit different functions, and even the same gene family may have various functions. Consequently, in this study, we found that the *GRF* genes contain diverse biological functions. Our comparative analysis suggests that the biological function of the *GRF* gene family is closely linked to protein structure, gene evolution or duplication events, and protein interactions.

Firstly, we detected a total of 24 distinct 3D structures of *GRF*s, indicating diverse biological functions among *Dendrobium*species. Secondly, gene evolution and duplication events also affect the biological function of *GRF* genes. For example, (i) *DoGRF13* and *DcGRF3* show homologous relationships with *A. thaliana*, *O. sativa*, *V. planifolia*, and each other; (ii) *DoGRF7*, *DoGRF2*, *DoGRF8* and *DcGRF17*, *DcGRF2*, *DcGRF15* show homologous relationships with *O. sativa*, *V. planifolia*, and each other; (iii) Collinearity analysis detected 3 pairs of *GRF*s with close relationships among *Dendrobium*species (*DcGRF17-DoGRF7*, *DcGRF2-DoGRF2*, *DcGRF15-DoGRF8*). These results indicate that *GRF*s have a conserved evolutionary history within the *Dendrobium* genus. However, *GRF*s also show a diversified evolutionary history among orchid species and other plant lineages. For example, (i) *Vpl04Ag09642* of *V. planifolia* has homologous pairs with two *DoGRF*s (*DoGRF8* and *DoGRF2*); (ii) *AT1G78300* of *A. thaliana* has homologous pairs with two *DcGRF*s (*DcGRF11* and *DcGRF7*). Considering the conserved evolutionary history within the *Dendrobium* genus but diversified evolutionary history among different plant lineages,
we suggest that gene evolution and duplication events affect the biological function of *GRF* genes.

Thirdly, interactions between different *GRF* proteins also affect their biological functions. We detected a total of 233 interactions between 15 *GRF* proteins and 50 related proteins. Among them, three *DoGRFs* (*DoGRF12*, *DoGRF16*, and *DoGRF17*) have the closest interaction relationship with PBSO (oxygen-evolving enhancer protein 1, chloroplastic). CMDH (Malate dehydrogenase) is also present in related proteins, indicating a possible correlation between *GRFs* and photosynthesis in *Dendrobium* s. Therefore, we suggest that the biological function of the *GRF* gene family is closely related to protein structure, gene evolution or duplication events, and protein interactions.

# CONCLUSIONS

In the current investigation, we identified and verified a total of 19 *DoGRFs* and 18 *DcGRFs* in the genomes of *D. officinale* and *D. chrysotoxum*, respectively. The *DoGRFs* and *DcGRFs* are distributed randomly across various chromosomes and classified into six subfamilies. We conducted a comprehensive analysis of gene structure, molecular evolution, interaction networks, and expression profiles to gain insights into the evolution of *GRF* genes in studied *Dendrobium* species. Our findings provide important information on the evolution of *GRF* genes in *Dendrobium* species.

# ACKNOWLEDGEMENTS

We would like to express our gratitude to the School of Life and Health Sciences at Huzhou College and the College of Life Sciences at Nanjing Normal University for supplying the necessary equipment for this study. Additionally, we would like to extend our appreciation to the Jiangsu Provincial Engineering Research Center for Technical Industrialization for *Dendrobium* s for providing valuable technical assistance.

## Funding

Our work was funded by the National Natural Science Foundation of China (Grant No. 31702000, 32070353, and 31900268), the Forestry Science and Technology Innovation and Promotion Project of Jiangsu Province (LYKJ[2021]12), and the Foundation of Key Laboratory of National Forestry and Grassland Administration for Orchid Conservation and Utilization (OC202102). The funders had no role in study design, data collection and analysis, decision to publish, or preparation of the manuscript.

## Grant Disclosures

The following grant information was disclosed by the authors:
National Natural Science Foundation of China: 31702000, 32070353, 31900268.
Forestry Science and Technology Innovation and Promotion Project of Jiangsu Province: LYKJ[2021]12.

Foundation of Key Laboratory of National Forestry and Grassland Administration for Orchid Conservation and Utilization: OC202102.

## Competing Interests

The authors declare there are no competing interests.

## Author Contributions

- Shuying Zhu performed the experiments, analyzed the data, prepared figures and/or tables, authored or reviewed drafts of the article, and approved the final draft.
- Hongman Wang conceived and designed the experiments, prepared figures and/or tables, authored or reviewed drafts of the article, and approved the final draft.
- Qiqian Xue performed the experiments, analyzed the data, prepared figures and/or tables, and approved the final draft.
- Huasong Zou analyzed the data, prepared figures and/or tables, authored or reviewed drafts of the article, and approved the final draft.
- Wei Liu analyzed the data, prepared figures and/or tables, and approved the final draft.
- Qingyun Xue analyzed the data, prepared figures and/or tables, and approved the final draft.
- Xiao-Yu Ding conceived and designed the experiments, prepared figures and/or tables, authored or reviewed drafts of the article, and approved the final draft.

## Data Availability

The raw data is available in the Supplemental Files.

## Supplemental Information

Supplemental information for this article can be found online at http://dx.doi.org/10.7717/peerj.16644#supplemental-information.

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
