# Peer review of "Genome-wide identification and expression analysis of growth-regulating factors in Dendrobium officinale and Dendrobium chrysotoxum"

_PeerJ, doi:10.7717/peerj.16644_

## Round 0.1 · original submission · Major Revisions

Please address the reviewers' comments to change the manuscript, and submit it to the system.

Reviewer 1 ·

Basic reporting

The manuscript titled "Genome-wide Identification and Expression Analysis of Growth-Regulating Factors in Dendrobium officinale and Dendrobium chrysotoxum" presents a comprehensive investigation of the growth-regulating factors (GRFs) family in two species of the genus Dendrobium. The authors successfully identified all the GRF genes in these species and explored their genomic distribution patterns, evolutionary mechanisms, and functions. The findings of this study are highly intriguing and contribute significantly to our understanding of growth regulatory mechanisms in Dendrobium species. While the analysis and results presented in the article are generally sound, I have identified a few areas that require attention, as outlined below:

1. In the Introduction section, it would be beneficial to provide a more detailed description of the two species, including any similarities and differences in their growth habits (lines 93-94).
2. Line 86: Please provide the scientific name of the taxa. Please note that the scientific name of plants should always include the author name when it is first mentioned in the main text.
3. Line 129: Could you clarify whether the sequence used here refers to the amino acid sequence or the nucleotide sequence?
4. Line 130: The analysis was performed using MEGA7 (Kumar, Stecher & Tamura, 2016).
5. Line 132: Please remove the reference to MEGA7.
6. Line 198: Chromosome 10 (Chr10) exhibited the highest number of DcGRF genes (Table 2).
7. Line 163: Please provide the reference for the Stringtie software.
8. Line 173: Please provide the reference for the Snapgene software.
9. Line 209: How many pairs of GRF genes were identified in the two Dendrobium species?
10. Line 210: ...together. Notably, we identified 8 pairs of orthologous genes with a ...
11. Line 212: Please provide additional details on the two subfamilies for the readers' better understanding.
12. Lines 252-253: Please include information on the findings in other species as well.
13. Figure 1: Please explain the numbers on the branches. Additionally, why do some branches lack bootstrap values?
14. Figure 4: What is the significance of the red lines?

Experimental design

no comment

Validity of the findings

no comment

·

Basic reporting

In this study, the GRF gene family in Dendrobium officinale and Dendrobium chrysotoxum were identified and analyzed from the aspects of evolution and function by using bioinformatics methods and experimental techniques. These findings provide valuable information for further investigations into the evolution and function of GRF genes in D. officinale and D. chrysotoxum. But there are still some problems in the manuscript. Following are some revisions.
1. The fonts in the figures are not uniform and including Times new roman, arial, and decorated letter at the same time. Please unify the font.
2. The gene names that appear in the picture should be in italics, such as Figure 6B.
3. In Figure 2, it is recommended to supplement the detailed information of each conserved motif sequence in the motif analysis, and the sequence logo can be supplemented as the supplementary figure. And the coordinates at the bottom of the exon-intron structures of GRF genes in D. officinale and D. chrysotoxum were suggested to be complete.
4. Line 28, ‘also for its great medicinal values’ should be ‘for its significant medicinal value.’
5. Line 32, ‘GRF family genes from Dendrobium officinale and Dendrobium chrysotoxum were identified by HMMER and BLAST.’ For ‘GRF’, best to not start a sentence with an abbreviation. Please also check throughout the manuscript.
6. Line 62-64, ‘Among them, the Growth-regulating factor (GRF), which has been proven to be involved in the growth and development of multiple plant organs, particularly in stems and leaves, plays an important role in plants.’ This sentence can be split into two sentences to reduce complexity.
7. Line 75, ‘cis’ should be italicized in ‘cis-acting’. Please correct and unify this format throughout the manuscript.
8. Line 146-149, ‘Ka’ and ‘Ks’ should be italicized. Please correct and unify this format throughout the manuscript.
9. Lines 157-165, Methods on expression files should be described in more detail.
10. Line 240-282, as a result, the section ‘Gene duplication and syntenic analysis of DoGRFs and DcGRFs’ is too long. It can be split into two results to reduce length.
11. Line 359, In the sentence, ‘For instance, Hewezi et al. (Hewezi et al., 2012) focused on the study of GRFs in A. thaliana and found that highly expressed AtGRF1 and AtGRF3 in roots had a balanced expression that affected root growth’, ‘Hewezi et al.’ should only be used once. Line 398, same situation, ‘Pajoro et al.’ should only be used once. Please correct and check throughout the manuscript.
12. Line 432-434, ‘Based on our comparative analysis, we suggest that the biological function of the GRF gene family is closely related to protein structure, gene evolution or duplication events, and protein interactions’ would be clearer as ‘Our comparative analysis suggests that the biological function of the GRF gene family is closely linked to protein structure, gene evolution or duplication events, and protein interactions.’

Experimental design

1. The manuscript analyzes sequences from the family of growth regulators in two Dendrobium species. The topic is exciting and novel and should allow inferences about the evolutionary processes associated with the functional differentiation of such sequences. However, its main limitation is that only two species were analyzed, which is about inferring processes for the entire genus, with more than 213 species with global distribution.
2. The authors should estimate the divergence time of duplicated GRF gene pairs using their Ks values.
3. In ‘Expression patterns of GRFs in different tissues and under MeJA treatments’, the focus of research on GRF function in Dendrobiums was more focused on the elongation of Dendrobiums’ stems, why not validate the stems specific expression patterns of these GRF genes via qRT-PCR. Besides, I also recommend validating the expression patters of the transcriptomic analysis with further experiments (e.g. ISH or FISH).

Validity of the findings

no comment

Additional comments

no comment

Reviewer 3 ·

Basic reporting

This study is involved in the evolution and biological function of GRF genes in Dendrobium officinale and Dendrobium chrysotoxum. A total of 19 DoGRFs and 18 DcGRFs were identified, respectively. And the evolution of GRF gene family in Dendrobiums is conserved. Through cis-acting element analysis, transcriptome analysis, protein 3D structure and protein interaction network construction, the insights into the biological function of GRF genes were gained in studied Dendrobium species. These findings provide important information on the evolution of GRF genes in Dendrobium species. But there are still some problems in the manuscript.

Experimental design

1. The authors have performed cross-species analysis and it would be desirable to provide a phylogenetic tree that could summarize the changes of these gene families in each node.
2. The tissue-specific expression of GRF genes in D. officinale and D. chrysotoxum was analyzed and discussed in the manuscript, and it is suggested to further explore the cell location of GRF genes function from the microscopic point of view through subcellular localization prediction.
3. It is recommended to supplement the details of each protein in ‘Figure 7. Protein-protein interaction (PPI) networks of GRF proteins in D. officinale’ as a supplement table. In addition, I would recommend validating the protein-protein interactions analysis with some molecular biology approaches, such as Y2H, BiFC and FRET.
4. The version of all the software tools in the ‘Materials and Methods’ section should be provided.
5. Before using abbreviations make sure you have firstly introduced the full name. For example, line 32, ‘Dendrobium officinale’ and ‘Dendrobium chrysotoxum’. Please also check throughout the manuscript.
6. Line 38-40, the sentence ‘Sequence comparison analysis showed relatively conserved gene structures and motifs in the same subfamily members, which indicate that the evolution of GRF genes was conserved among Dendrobium species.’ would be clearer as ‘Sequence comparison analysis revealed relatively conserved gene structures and motifs among members of the same subfamily, indicating a conserved evolution of GRF genes within Dendrobium species.’
7. The online software and database used in the materials and methods shall provide the corresponding website address. For example, line 117, Pfam protein family database; line 120, Ensembl Plants Database; line 124, ExPASy software online. And ensure that the full names of online software and databases are complete. Please also checked throughout the manuscript.
8. Line 129-130, please provide the outgroups information in phylogenetic analysis.
9. Line 152, ‘1500bp’ should be ‘1500 bp’ (add a space). Please also checked throughout the manuscript.
10. The names of all reagents appearing in materials and methods shall be written in a standardized manner. For instance, line 170, ‘ddH2O’ should be ‘ddH2O’. Please also check throughout the manuscript.

Validity of the findings

11. As a sentence, the title of each discussion point should end with a period.
12. Line 358, ‘…phytohormone responses, growth and development regulation, and stress responses (Vercruyssen et al., 2015)’, ‘growth and development regulation’ should be ‘regulation of growth and development’.
13. Please check the reference format carefully, journal names should not be abbreviated. For example, in line 593-594 of the references ‘Kumar S, Stecher G, Tamura K. 2016. MEGA7: Molecular Evolutionary Genetics Analysis Version 7.0 for Bigger Datasets. Mol Biol Evol. 33:1870-4’, ‘Mol Biol Evol.’ Should be ‘Molecular Biology and Evolution’.

Additional comments

The English language should be improved.

---

## Round 0.2 · Minor Revisions

Please modify it according to the reviewers' comments, then resubmit it.

Reviewer 1 ·

Basic reporting

This revision has been greatly improved overall. The changes and responses undertaken largely answer my earlier concerns. Here are a few issues that need to be clarified in the text by the authors.
1. “First, the GRF sequences of D. officinale, D. chrysotoxum, Phalaenopsis equestris” should be “First, the GRF amino acid sequences of D. officinale, D. chrysotoxum, Phalaenopsis equestris”

2. Line 138: “The resulting phylogenetic trees were constructed” should be “The phylogenetic trees were constructed”

3. Line 147: “value threshold = 1e-20 against the genome sequence of DNA” should be “value threshold = 1e-20 against the genome sequence of the two Dendrobium species”


4. Lines 172-173: The sentence “Then the clean reads were aligned to the D. officinale genome, and mapping by Hisat2 v2.2.1.” could be rewrite as “Then the clean reads were aligned and mapped to the D. officinale genome by Hisat2 v2.2.1.”?

5. Line 173: “And the data was sam to bam by SAMtools v1.14.” could be rewrite as “The sam data was converted to bam by SAMtools v1.14.”?

6. “Notably, we identified 8 pairs of orthologous genes with a close relationship. We identified 8 pairs of orthologous genes based on a Bootstrap value > 90, such as DoGRF14 and DcGRF12.” should be rewrite, e.g., “Notably, we identified 8 pairs of orthologous genes with a close relationship (Bootstrap value > 90), such as DoGRF14 and DcGRF12.”

7. In the legend of Figure 2, please add information of bootstrap value that showed in Fig. 2.

Experimental design

no comments

Validity of the findings

no comments

·

Basic reporting

no comment

Experimental design

no comment

Validity of the findings

no comment

Additional comments

no comment

Reviewer 3 ·

Basic reporting

no comment

Experimental design

no comment

Validity of the findings

no comment

Additional comments

no comment

---

## Round 0.3 · accepted · Accept

Congratulations! Thanks for your work to the journal.

Reviewer 1 ·

Basic reporting

This revision has been greatly improved overall. I have no further comments.

Experimental design

no comment

Validity of the findings

no comment

Additional comments

no comment